# Therapeutic timing limitations of postnatal darbepoetin in a valproic acid rat model of Autism Spectrum Disorder

Ömer Yusuf İpek[1,2]*, Tuğba Kirboğa[1], Ercan Babur[1], Nurcan Dursun[1], Cem Süer[1]

**1** Department of Physiology, Faculty of Medicine, Erciyes University, Kayseri, Turkey, **2** Department of Physiology, Faculty of Medicine, Ahi Evran University, Kırşehir, Turkey

* omer.ipek@ahievran.edu.tr

**Citation:** İpek ÖY, Kirboğa T, Babur E, Dursun N, Süer C (2025) Therapeutic timing limitations of postnatal darbepoetin in a valproic acid rat model of Autism Spectrum Disorder. PLoS One 20(11): e0337294. https://doi.org/10.1371/journal.pone.0337294

## Abstract

Autism spectrum disorder (ASD) arises from complex genetic and environmental factors that disrupt neural development during early brain maturation. Erythropoietin (EPO) has been studied for its neuroprotective effects and more recently for its potential to influence neurodevelopment in early postnatal ASD models. However, ASD is not typically diagnosed in humans until 2–3 years of age, a stage well beyond early postnatal development. To address this timing gap, we administered darbepoetin alfa, a long-acting EPO analogue, to valproic acid-exposed rats beginning at postnatal day 21 for five consecutive days, and assessed ASD-relevant social and cognitive behaviors. Behavioral assessments using the three-chamber test and Morris Water Maze revealed no significant improvements in ASD-relevant behaviors despite clear systemic activity, as evidenced by substantial hematocrit elevation (~70%). Our findings suggest the therapeutic window for EPO analogues may close before the post-diagnostic period, highlighting a critical translational challenge: interventions effective in early neonatal windows may not retain efficacy at clinically accessible diagnostic stages. The pronounced hematological response further precludes testing whether higher doses could compensate for delayed timing, though non-erythropoietic derivatives may circumvent this limitation in future studies.

## Introduction

Autism spectrum disorder (ASD) is a neurodevelopmental condition characterized by impairments in social interaction and communication, alongside restricted and repetitive behaviors [1,2]. Recent systematic reviews estimate a global prevalence of 1–4%, with notable variation across geography and socioeconomic context [3]. The etiology of ASD is multifactorial, reflecting interactions between genetic susceptibility and environmental influences during sensitive periods of brain maturation [4]. One of the most well-documented environmental risk factors is prenatal exposure to valproic acid (VPA), which has been shown to increase the incidence of ASD [5,6].

**Data availability statement:** The EthoVision export files (raw behavioral data), hematological parameters and Python-generated statistical output logs are publicly available on a Zenodo repository (https://doi.org/10.5281/zenodo.15727028).

**Funding:** This manuscript is based on the first author's PhD thesis project, funded by Erciyes University Scientific Research Projects Coordination Unit. The funders had no role in study design, data collection and analysis, decision to publish, or preparation of the manuscript.

**Competing interests:** The authors have declared that no competing interests exist.

The prenatal VPA model is therefore widely used to study ASD-relevant mechanisms and interventions. It provides strong construct validity by replicating a known etiological exposure and reliably yields social and cognitive alterations within a defined neurodevelopmental context [7,8]. In line with established protocols, we administered 500 mg/kg VPA at embryonic day 12.5, a critical period for cortical neurogenesis [8]. We employed Wistar albino rats, in which the VPA model is extensively characterized and social behaviors are well documented. Importantly, ASD-like behavioral alterations in this model are reported to be more robust and reproducible in male offspring than in females [9,10]. To reduce variability and enhance comparability with prior work, we therefore restricted our cohort to males. We note that as an environmental-insult model, VPA does not represent the broader heterogeneity of ASD, and all the behavioral components of autism in humans can't be replicated in rats. Despite these limitations, this model provides a robust and relevant neurodevelopmental framework for testing the timing of therapeutic interventions.

Erythropoietin (EPO), a glycoprotein cytokine traditionally recognized for its hematopoietic role, also exerts potent neuroprotective and anti-inflammatory effects in the central nervous system [11,12]. These functions intersect with neurodevelopmental processes relevant to ASD, including neuroinflammation and aberrant synaptic pruning [13–16]. In rodent models of ASD, early postnatal EPO treatment has shown promise in improving behavioral outcomes, suggesting that targeting this pathway may be a viable therapeutic strategy [17].

A major challenge, however, lies in translation. ASD can be diagnosed around 2–3 years of age, by which time many sensitive windows for brain maturation have passed [18,19]. This timing makes interventions that require immediate neonatal administration clinically difficult to implement. To address this translational gap, we aimed to test whether an EPO-related therapeutic could be effective when administered in a later, more clinically relevant "post-diagnostic" window. We chose postnatal day (P) 21 in rats, a juvenile stage that corresponds to a human toddler of approximately two to three years of age [19]. This alignment is based on the convergence of multiple key neurodevelopmental milestones, including the brain reaching 90–95% of its adult weight, alongside peaks in both synaptic density and the rate of myelination [19]. Crucially, this window is also characterized by the onset of intense, activity-dependent synaptic pruning, a major event that refines neural circuits into their mature state [20]. This process is known to be disrupted in ASD, and can be modulated by EPO receptor signaling [13,14]. This window is therefore a particularly relevant time for intervention, as it allows for targeting a core pathological process during a period of active and dynamic circuit maturation.

We selected darbepoetin alfa (DPO), an EPO analogue with demonstrated neuroprotective effects in both animal and clinical studies, paralleling those reported for EPO [21,22]. DPO's additional sialic acid residues prolong its half-life approximately threefold, supporting sustained receptor engagement [23]. This property, combined with prior neuroprotective evidence and its clinical availability, made it a translationally relevant analogue to test in this neurodevelopmental context. In clinical hematopoietic use, less frequent dosing is standard for both EPO and DPO in humans [24,25];

however, neuroprotective regimens are not yet standardized, and to our knowledge this is the first study to evaluate darbepoetin in a neurodevelopmental model. We therefore designed a protocol of 10 µg/kg/day for five consecutive days (P21-25), based on the commonly cited ~1:200 EPO:DPO bioequivalence ratio [26] and aligned with the 2000 IU/kg/day × 5 postnatal EPO protocol previously applied in a VPA model [17]. Although this conversion ratio can vary substantially by species, endpoint and dosing schedule [23,26], our cumulative 50 µg/kg DPO dose matches that used by Cherian et al. with demonstrated neuroprotective efficacy [27], and remains conservative within EPO neuroprotection protocols that have employed cumulative doses up to 70,000 IU/kg [28,29]. Because elevations in hematocrit are recognized class effects of EPO analogues at neuroprotective doses [28], we monitored hematological parameters as predefined safety outcomes.

To characterize the effects of DPO treatment, we assessed social behavior at P30 using the three-chamber test. Spatial learning and cognitive flexibility were evaluated in the Morris Water Maze with reversal learning at P56-63. These tasks were selected because the three-chamber test robustly models social approach, a core ASD-relevant domain [9,30], while the Morris Water Maze provides a well-established index of hippocampal-dependent learning and cognitive flexibility in VPA models, serving as a neurodevelopmental indicator relevant to ASD pathology [10,31]. Beyond standard social preference metrics, we quantified alternation rate and immobility from the three-chamber test to capture additional dimensions of behavioral flexibility and exploratory drive relevant to restricted and repetitive behavior domains in ASD. Hematological parameters were measured at P21 and P28, before and after DPO treatment, to assess systemic effects and monitor safety during this developmental stage. The aim of this study was therefore to evaluate whether DPO, administered in a juvenile window corresponding to the human post-diagnostic period, could improve ASD-relevant behaviors in the VPA model, while also assessing its hematological effects as a safety consideration.

## Materials and methods

### Ethical approval

The animal study protocol was approved by the Erciyes University Animal Experiments Ethics Committee (Decision: 23/062, Date: 05.04.2023).

### Animals

A total of 32 female and 16 male Wistar albino rats were obtained from the Erciyes University Experimental Research and Application Center (Kayseri, Turkey) for breeding. Animals were housed under standard laboratory conditions (12 h light/dark cycle, 21 ± 2 °C, 50–60% humidity) with ad libitum access to food and water. Female rats were paired with males overnight, and successful mating was confirmed the following morning by the presence of sperm in vaginal smears, designated as gestational day 0.5 (GD0.5). Only male offspring were included in the study. Surplus pups were returned to the Erciyes University animal facility in accordance with the ethics committee's protocols.

Animals included in the study (n = 64) were monitored daily for health and behavior. The protocol defined humane endpoint criteria, including veterinary recommendation, markedly reduced responsiveness, or marked locomotor impairment, but no animals met these conditions during the study, and no unexpected morbidity or mortality occurred.

The study design included planned euthanasia at the conclusion of behavioral testing, which was reviewed and approved as part of the protocol. All animals were euthanized at the planned experimental endpoint postnatal day 63 (P63) under deep urethane anesthesia (i.p., 1.2 g/kg), followed by decapitation, in accordance with institutional guidelines. All procedures were conducted by certified personnel for animal handling and euthanasia.

### Experiment design

On GD12.5, 20 pregnant dams were weighed and randomly assigned to receive a total intraperitoneal (i.p.) dose of 500 mg/kg valproic acid (VPA; Sigma-Aldrich, P4543) dissolved in saline (4 mL/kg volume), administered in two 250 mg/kg

injections with a 5-minute interval between doses to reduce acute toxicity. The remaining 12 pregnant females received an equal volume of sterile saline (4 mL/kg). VPA exposure was used to induce an established rodent model of autism spectrum disorder (ASD). Dams were allowed to deliver naturally and raise their litters under standard housing conditions.

To ensure precise timing of VPA administration on gestational day 12.5 (GD12.5), a two-step verification protocol was implemented. First, male breeders were removed immediately upon detection of sperm-positive vaginal smears, minimizing the risk of delayed fertilization that could result in premature VPA exposure. Second, birth dates were tracked to identify any discrepancies between assumed and actual gestational timing. All litters were delivered on post-conception days 21–23, confirming consistent dating and largely ruling out early or late fertilization. This protocol ensured accurate gestational staging and uniform timing of VPA administration across all experimental subjects.

At P21, male offspring (body weight ~40±4 g) were weaned and randomly assigned to four experimental groups (n = 16 each) based on prenatal exposure to either VPA or saline. Within each group, animals received postnatal darbepoetin alfa (10 µg/kg; Aranesp®, Amgen Inc.) or sterile saline (1 mL/kg), administered intraperitoneally once daily for five consecutive days (P21-P25). A schematic of the experimental timeline and group assignments is shown in Fig 1.

## Behavioral tests

Behavioral testing was conducted between postnatal days 30 and 63. All 16 male offspring per group were subjected to behavioral assessment. All procedures were performed during the light phase in a temperature-controlled, minimally disturbed environment.

## Three-chamber social interaction test

Social behavior was evaluated using a three-chamber apparatus (120 cm long, 60 cm wide, 35 cm high) consisting of two side compartments and a central zone (each 40×60 cm), separated by transparent walls with sliding doors. Wire cages (~8.5×8.5 cm) were placed diagonally in each side chamber. For behavioral analysis, interaction zones were defined as 10×10 cm squares centered around each cage. Zone entry was determined by nose-point position to capture active

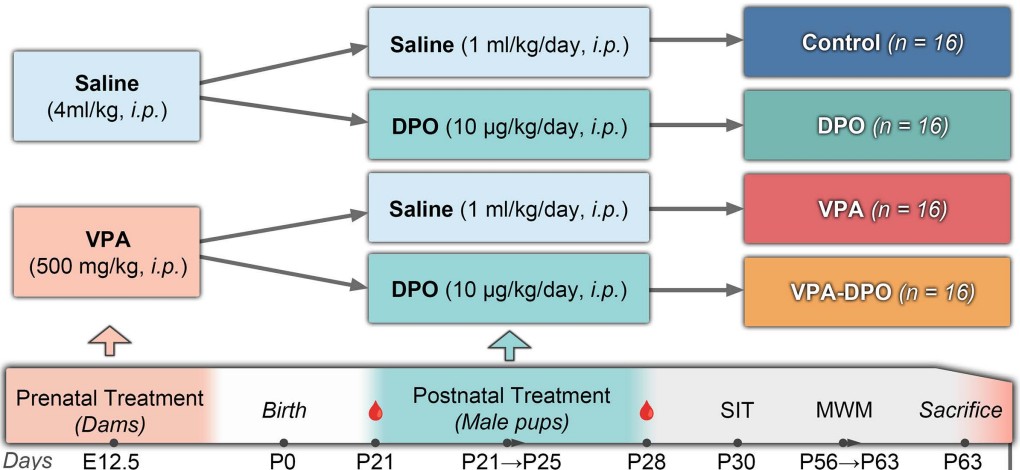

**Fig 1. Experimental design.** Pregnant dams received valproic acid (VPA; 500 mg/kg, i.p.) or saline on embryonic day 12.5 (E12.5). Male offspring (n = 16 per group) received postnatal saline or darbepoetin alfa (DPO; 10 µg/kg/day, i.p.) once daily on postnatal days 21-25 (P21-P25). Blood samples were collected at P21 and P28 from a subset of animals (n = 6 per group). Behavioral testing included the three-chamber social interaction test (SIT) at P30 and the Morris Water Maze (MWM) with reversal learning at P56-P63. All animals were euthanized on P63. Abbreviations: E, embryonic day; P, postnatal day; i.p., intraperitoneal.

investigative behavior. Zones were detected and tracked automatically using EthoVision XT software. Testing spanned two consecutive days:

**Day 1 (Habituation)** Each rat was individually placed in the center chamber and allowed to freely explore all three sections for 10 minutes.

**Day 2 (Sociability & Social novelty)**

- **Sociability phase:** An unfamiliar, same-sex, age-matched conspecific (Stranger 1) was placed inside a wire cage in one side chamber, while the other side chamber contained an identical empty cage. The subject rat was placed in the center chamber and allowed to explore freely for 10 minutes.

- **Social novelty phase:** Following the sociability phase, the subject rat was returned to its home cage for a 5-minute inter-trial interval. During this time, the apparatus and cages were cleaned with 70% ethanol to minimize olfactory cues. The now-familiar conspecific (Stranger 1) was transferred to the cage in the opposite chamber (which had been empty). A second, novel conspecific (Stranger 2) was then placed into the cage that originally held Stranger 1. The subject rat was then returned to the center chamber and allowed to explore for a final 10-minute session, where it could choose between the familiar and the novel Stranger 2. The apparatus and cages were cleaned with 70% ethanol between animals to eliminate olfactory interference.

All sessions were recorded using a ceiling-mounted video system and analyzed using Noldus EthoVision XT. Time spent in each interaction zone was used to calculate the Social Preference Index (SPI) as:

$$SPI = (Time\ with\ stranger - Time\ with\ empty\ cage)/Total\ time\ in\ both\ zones$$

Values range from −1 (complete non-social preference) to +1 (complete social preference), with 0 indicating no preference.

Although alternation rate is more commonly used in Y-maze and T-maze paradigms to assess spatial working memory and exploratory behavior, we adapted this metric to the three-chamber test to examine dynamic switching between social targets. This exploratory measure was intended to capture subtle shifts in social engagement and behavioral flexibility beyond standard preference indices. Alternation was defined as a consecutive transition from one interaction zone to the other (e.g., from the social to the non-social zone or vice versa). It was calculated as:

$$Alternation\ Rate = Number\ of\ Alternations/(Social\ Count + NonSocial\ Count - 1)$$

where the denominator reflects the maximum number of alternations possible, given the total number of zone entries. Reduced alternation may indicate decreased exploratory motivation, social disengagement, or increased anxiety-driven zone perseveration, all of which are relevant to modeling ASD-related phenotypes.

## Morris Water Maze (MWM) with reversal learning

Spatial learning and cognitive flexibility were evaluated using a circular water maze (120 cm diameter, 50 cm deep) filled with opaque water maintained at $22 \pm 2°C$. A submerged escape platform (10 cm diameter) was hidden 1 cm beneath the surface in one of four quadrants, with prominent distal visual cues placed around the testing room.

The procedure consisted of:

- Days 1–4 (Initial learning): Each rat underwent four trials per day, starting from different quadrants in a clockwise rotation, avoiding the platform-containing quadrant. Animals were given 1 minute to find the platform. When an animal reached the platform and remained on it for one second, trial was concluded with the time saved as "Escape Latency". 'First-entry latency' was defined as the time from trial start until the animal's initial contact with the platform, regardless of whether it remained there.

- Day 5 (Probe trial): The platform was removed to assess memory retention of the trained location. Animals were allowed to explore for 2 minutes in the water maze. The 'platform zone' was defined as the circular area (10 cm diameter) previously occupied by the platform, and time spent in this zone was recorded. A 'platform crossing' was counted when the animal's center point passed through this zone.

- Days 6–8 (Reversal learning): The platform was relocated to the quadrant opposite the original location. Rats underwent four trials per day following the same procedure as during the initial learning phase to evaluate cognitive flexibility. Each trial had a 1-minute maximum duration, and escape latency was recorded as described.

Swim paths, escape latency, platform crossings and time spent in each quadrant were recorded using a ceiling-mounted video system and analyzed using EthoVision XT.

The thigmotaxis zone was operationally defined as a ring-shaped area spanning the outer 10% of the maze radius. This zone was used to quantify wall-oriented swimming behavior, and was automatically tracked for duration and distance using the EthoVision system.

"Meander" was defined as the average angular change per unit distance traveled (degrees/cm), quantifying path curvature. Elevated meander values reflect increased search irregularity and reduced spatial focus.

"Heading deviation" was defined as the average angular difference between the animal's movement vector and the direct line to the platform, calculated at each time point. Higher values indicate less accurate orientation. While meander reflects overall path curvature, heading deviation provides a moment-to-moment measure of navigational accuracy, capturing subtle deficits in spatial targeting, especially on shorter or more goal-directed swims.

### Blood collection

Tail vein blood was collected on postnatal day 21 (P21) and postnatal day 28 (P28) to assess hematological status (n = 6 per group). Due to an equipment failure, P21 samples from the VPA group could not be analyzed. On P28, samples were successfully collected and analyzed from all four experimental groups: Control, DPO, VPA, and VPA-DPO. Animals were gently restrained under a soft cotton cloth, and a small drop of blood was obtained from the tail vein for immediate analysis of hemoglobin and hematocrit values using a portable hemoglobin reader (Mission Ultra Hb, ACON Laboratories, Inc., San Diego, CA, USA). Hematocrit results are presented in the main text, while corresponding hemoglobin data are provided in the Supporting Information S4 File.

### Statistical analysis

Data distribution was assessed using the Shapiro-Wilk test, and homogeneity of variances with Levene's test. Depending on data normality, group comparisons were conducted using one-way ANOVA or Kruskal-Wallis tests. Post-hoc comparisons were performed where appropriate; Dunn's tests following Kruskal-Wallis were corrected using the Holm-Bonferroni method. Welch's ANOVA and Games-Howell post-hoc tests were included in the analysis pipeline and were triggered only for a single variable (hemoglobin levels at P21), where normality was met but variance equality was violated. Statistical significance was defined as $p < 0.05$.

For repeated measures across days in the MWM, three-way ANOVA was applied with Day, VPA and DPO as factors. Two-way ANOVAs were used to test the main effects of VPA and DPO, as well as their interaction. ANOVA was used for all two-way and three-way repeated-measures designs due to the lack of widely accepted non-parametric alternatives, its general robustness to assumption violations and the balanced group sizes in this study. Effect sizes for pairwise comparisons were reported as Cohen's d, calculated as the difference between group means divided by the pooled standard deviation. Effect sizes were interpreted as small ($d \geq 0.2$), medium ($d \geq 0.5$) and large ($d \geq 0.8$) based on conventional thresholds.

All analyses were conducted in Python 3.12 using standard scientific computing libraries. Statistical tests and effect size calculations were performed with SciPy 1.11.4, Pingouin 0.5.5 and Statsmodels 0.14.4, while Pandas 2.2.3, NumPy 1.26.4, Seaborn 0.13.2 and Matplotlib 3.8.4 were used for data handling and visualization [32–38].

## Results

### Social behavior

In the three-chamber test, VPA-exposed and VPA-DPO animals both showed significantly reduced social preference compared to controls ($p = 0.004$; *Cohen's d* = 1.1), consistent with ASD-like social deficits (Fig 2A). A two-way ANOVA (VPA×DPO) confirmed a significant main effect of VPA on social preference ($F = 13.3$, $p < 0.001$), with no significant effect of darbepoetin ($F = 0.06$, $p = 0.8$) or interaction between the factors ($F = 0.9$, $p = 0.35$).

The VPA group exhibited markedly elevated within-group variability in social preference scores (SD = 0.49), significantly greater than controls (SD = 0.13; Levene's test: $F = 6.2$, $p = 0.019$), indicating more heterogeneous behavioral responses.

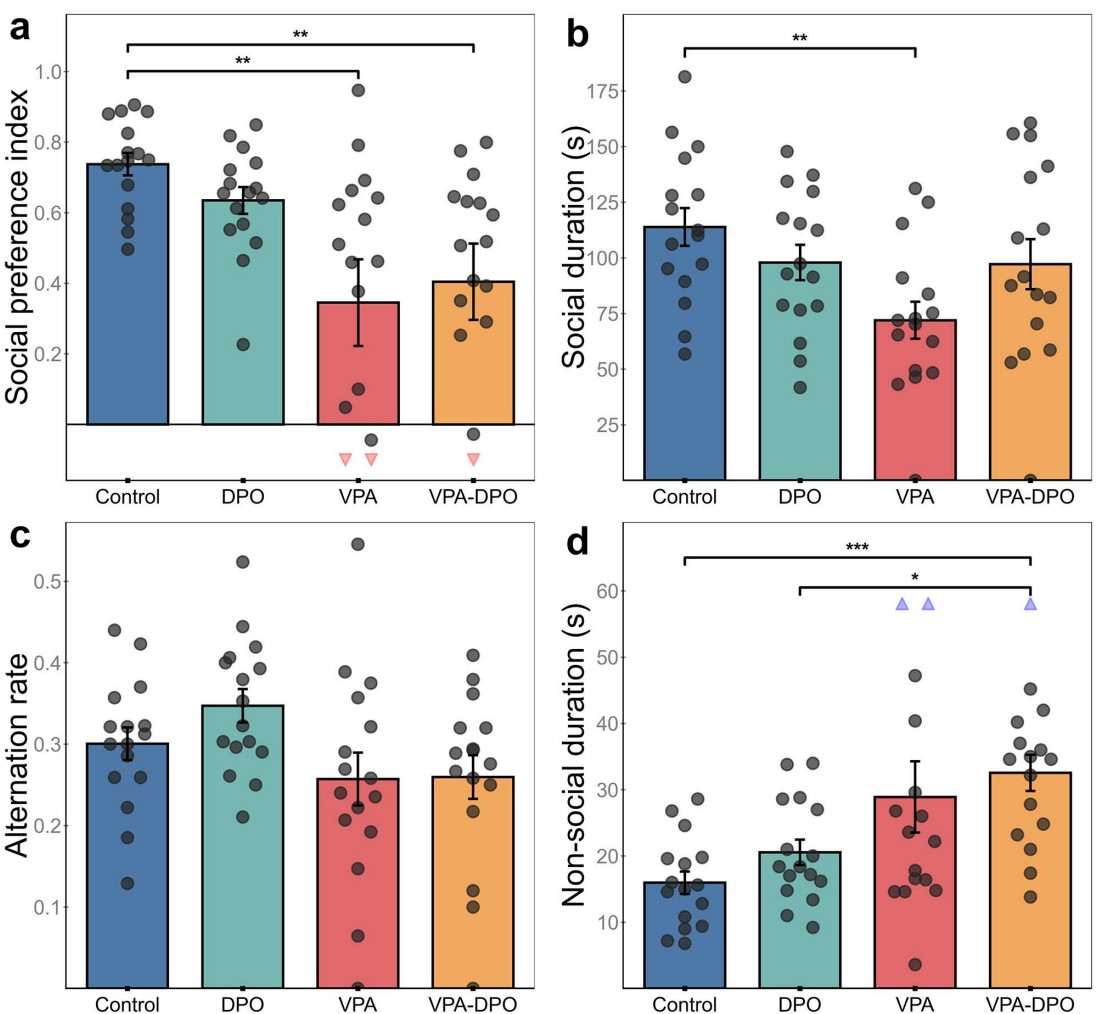

**Fig 2. Three-chamber test behavioral outcomes.** Individual data points are shown with bars representing group means ± SEM. **(a)** Social Preference Index was significantly reduced in VPA and VPA-DPO groups compared to Control ($p = 0.004$, $d = 1.1$). **(b)** Time spent in the social zone was significantly lower in the VPA group compared to Control ($p = 0.01$, $d = 1.25$). **(c)** Alternation rate revealed a main effect of VPA in two-way ANOVA ($F = 6.61$, $p = 0.013$), with no significant effect of DPO ($F = 0.93$, $p = 0.34$). **(d)** Time spent in the non-social zone was significantly elevated in VPA-DPO compared to Control ($p < 0.001$, $d = 1.8$) and DPO ($p = 0.036$, $d = 1.3$). Statistical comparisons: Kruskal-Wallis with Dunn's post hoc and Holm-Bonferroni correction **(a, d)**; one-way ANOVA with Tukey's HSD **(b, c)**. *$p < 0.05$, **$p < 0.01$, ***$p < 0.001$. For appropriate plot scaling, extreme data points were clamped and shown as downward red or upward blue triangles, but all values were included in statistical analyses.

In contrast, the DPO (SD = 0.15; Levene's test: $F = 0.08$, $p = 0.79$) and VPA-DPO groups (SD = 0.43; Levene's test: $F = 3.01$, $p = 0.09$) did not differ significantly from controls in variability.

A trend toward reduced alternation rate was observed in both the VPA and VPA-DPO groups (Fig 2C). Two-way ANOVA confirmed a significant main effect of VPA on alternation rate ($F = 6.6$, $p = 0.013$), with no significant effect of darbepoetin or interaction. Similarly, VPA exposure significantly increased immobility during the task ($F = 5.0$, $p = 0.029$), with no significant main effect of darbepoetin ($F = 1.2$, $p = 0.28$) or interaction ($F = 1.3$, $p = 0.26$).

Additionally, VPA exposure led to a significant increase in time spent in the non-social zone ($F = 14.5$, $p < 0.001$), increased number of visits to the non-social zone ($F = 5.9$, $p = 0.019$), reduced duration in the social zone ($F = 5.5$, $p = 0.022$), further suggesting reduced social engagement (Fig 2). Notably, a significant VPA×DPO interaction was observed for time spent in the social sniffing zone ($F = 5.1$, $p = 0.027$), indicating that DPO treatment may have partially normalized engagement with the social target. No other significant main effects or interactions were observed for these measures. Group-level average occupancy heatmaps are provided to visualize spatial engagement patterns during the social interaction test (Fig 3), complementing the quantitative results.

A subsequent social novelty trial conducted after a 5-minute break showed low engagement across groups and inconsistent behavioral patterns. Although a few parameters reached statistical significance (e.g., alternation or immobility), the overall data lacked coherence, possibly due to habituation effects or fatigue. As such, this phase was excluded from focused analysis, but raw data and summary statistics are available in the Zenodo repository. All statistical data for the three-chamber tests are provided in the Supporting Information S1 File.

### Spatial learning and memory

To evaluate spatial learning dynamics, we conducted three-way ANOVAs with Day, VPA exposure and DPO treatment as factors across key behavioral variables, while analyzing Days 1–4 and Days 6–8 separately. We also conducted daily group comparisons (one-way ANOVA or Kruskal-Wallis, as appropriate) and two-way ANOVAs (VPA×DPO) to assess group-level effects.

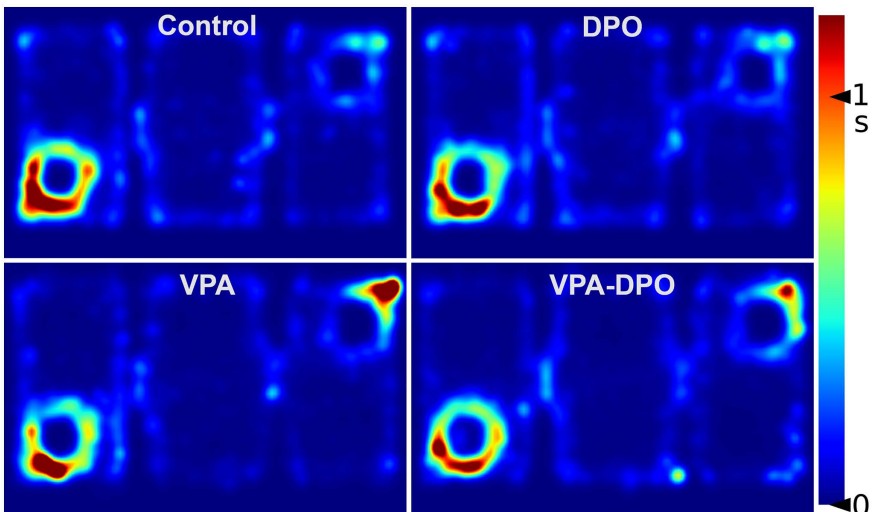

**Fig 3. Three-chamber test spatial patterns.** Group-level occupancy heatmaps generated from center-point tracking data depict spatial exploration during the sociability phase. Control and DPO groups focused on the social cage (bottom left), while VPA animals showed reduced social engagement. VPA and VPA-DPO groups spent more time near the non-social cage (top right). Each bin corresponds to ~1 cm²; the color scale indicates time spent per bin.

**Acquisition Phase (Days 1–4).** VPA-exposed animals exhibited worse learning outcomes, reflected by a significantly greater total distance swum across training days ($F = 8.9$, $p = 0.003$). DPO treatment also increased total distance ($F = 7.7$, $p = 0.006$), and a significant VPA×DPO interaction ($F = 4.2$, $p = 0.042$), indicating a combined effect (Fig 4C). A similar pattern was observed in escape latency (Fig 4A), where both VPA ($F = 4.7$, $p = 0.032$) and DPO ($F = 4.1$, $p = 0.043$) had significantly delayed escape latencies, and the VPA×DPO interaction further prolonged it ($F = 5.7$, $p = 0.017$). Interestingly, while VPA animals significantly took longer to reach the platform for the first time ($F = 5.0$, $p = 0.026$), DPO-treated animals did not (Fig 4B). Their first-entry latencies were slightly shorter than controls, though this difference was not statistically significant ($F = 0.01$, $p = 0.92$), indicating no evidence of impairment. Both total distance and escape latency significantly decreased across days, reflecting overall task adaptation.

In terms of spatial strategy, VPA animals showed disrupted search patterns, with significantly increased heading deviation ($F = 14.8$, $p < 0.001$) and meander ($F = 6.90$, $p = 0.009$), indicating less goal-directed navigation (Fig 4F). A strong VPA×DPO interaction was also found in heading deviation ($F = 14.4$, $p < 0.001$), indicating a further increase in path deviation. DPO alone did not significantly affect heading direction or meander.

Thigmotaxis, which represents the time spent close to the outer wall as a percentage, was markedly elevated in both VPA ($F = 13.1$, $p < 0.001$) and DPO ($F = 22.4$, $p < 0.0001$) groups. The VPA×DPO interaction also had a significant further effect on thigmotaxis ($F = 7.4$, $p = 0.007$), increasing it beyond either factor alone (Fig 4E).

**Probe Test (Day 5).** On the probe day, VPA-exposed animals showed consistent signs of impaired spatial memory. In two-way ANOVAs, they exhibited significantly longer latencies to first reach the former platform location ($F = 4.2$, $p = 0.044$), reduced time spent within that exact zone ($F = 16.45$, $p < 0.001$) and fewer direct platform crossings ($F = 4.8$, $p = 0.033$). Neither DPO alone nor the VPA×DPO interaction significantly altered these measures (Fig 5).

Group comparisons also supported these findings: VPA animals spent significantly less time in the former platform zone compared to Control ($p = 0.049$, $d = 1.04$), and VPA-DPO animals spent even less time than both Control ($p = 0.01$, $d = 1.2$) and DPO-treated animals ($p = 0.024$, $d = 1.0$).

No significant effects were observed for velocity, heading deviation, meander, total distance, immobility, or thigmotaxis (Fig 5D) in either two-way ANOVAs or group comparisons.

**Reversal Learning (Days 6–8).** Spatial performance impairments in VPA-exposed animals were not only sustained but appeared more pronounced during the reversal phase. Three-way ANOVAs showed that VPA significantly increased escape latency ($F = 9.0$, $p = 0.003$), delayed first contact with the platform ($F = 14.3$, $p < 0.001$), and resulted in longer total distance swum ($F = 12.8$, $p < 0.001$). DPO alone did not significantly affect these measures, although a significant VPA×DPO interaction was observed for total distance ($F = 12.1$, $p < 0.001$), reflecting reduced swim distance in the combined group (Fig 4).

Spatial strategy impairments in VPA animals persisted during reversal learning. VPA exposure significantly increased heading deviation ($F = 34.2$, $p < 0.0001$), indicating continued disruption in path directionality (Fig 4F). Meander was also elevated in VPA animals ($F = 5.0$, $p = 0.027$), and a significant VPA×DPO interaction was observed ($F = 7.3$, $p = 0.007$), suggesting a compounded effect on path curvature. DPO alone did not significantly affect either metric. Thigmotaxis remained elevated in both VPA ($F = 11.1$, $p = 0.001$) and DPO ($F = 10.2$, $p = 0.002$) groups, with no significant interaction between them (Fig 4E). As expected, all three parameters improved over the days, indicating adaptation.

Swim velocity exhibited a clear day-dependent shift, with a significant main effect of Day ($F = 11.7$, $p < 0.0001$), reflecting a general decline in swim speed over time (Fig 4D). During acquisition (Days 1–4), VPA animals also swam significantly faster than controls ($F = 5.9$, $p = 0.016$). In contrast, during reversal (Days 6–8), no significant Day effect emerged ($F = 1.1$, $p = 0.350$), and VPA animals instead swam significantly slower than controls ($F = 4.1$, $p = 0.045$).

Day-specific comparisons generally yielded patterns consistent with the three-way ANOVA results, though with lower statistical power and increased variability across individual days. Due to the large volume of data, only the most relevant

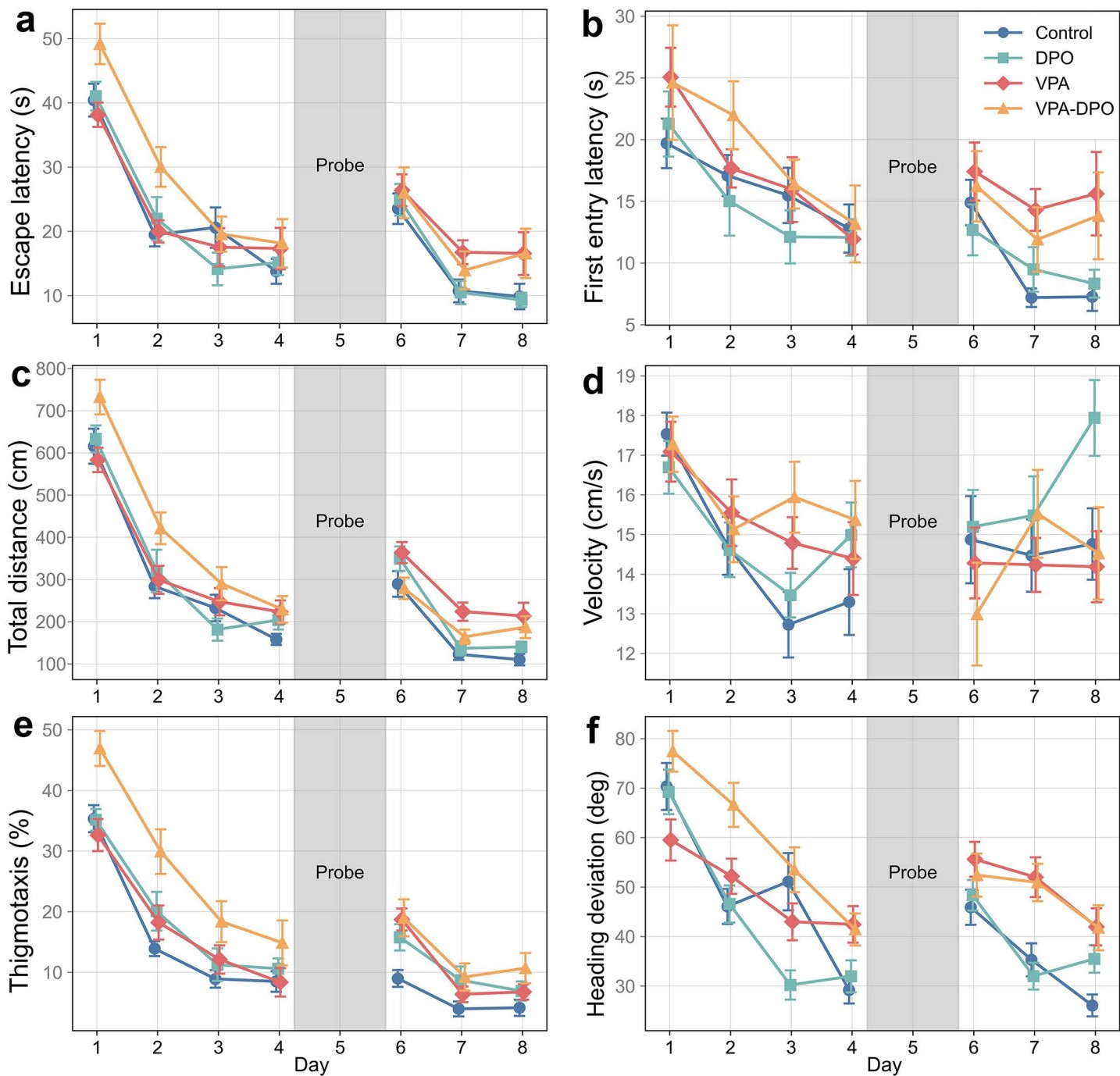

**Fig 4. Morris water maze performance across acquisition and reversal phases.** Each panel presents group means ± SEM across eight testing days. Days 1-4 represent spatial acquisition, Day 5 is the probe trial (gray shaded), and Days 6-8 assess reversal learning. **(a)** VPA exposure led to prolonged escape times during both learning phases, indicating impaired spatial memory. **(b)** VPA rats showed delayed initial contact with the platform, especially during reversal, suggesting deficits in spatial targeting. **(c)** VPA groups consistently traveled longer total distances, reflecting less efficient search strategies. **(d)** Swim velocity exhibited day-dependent changes, with VPA animals swimming faster during acquisition but slower during reversal, potentially reflecting altered search strategy. **(e)** Time spent near the wall (thigmotaxis) was elevated in both VPA and DPO groups, suggesting anxiety-like behavior or altered motivation. **(f)** VPA animals showed greater heading deviation, consistent with impaired spatial orientation.

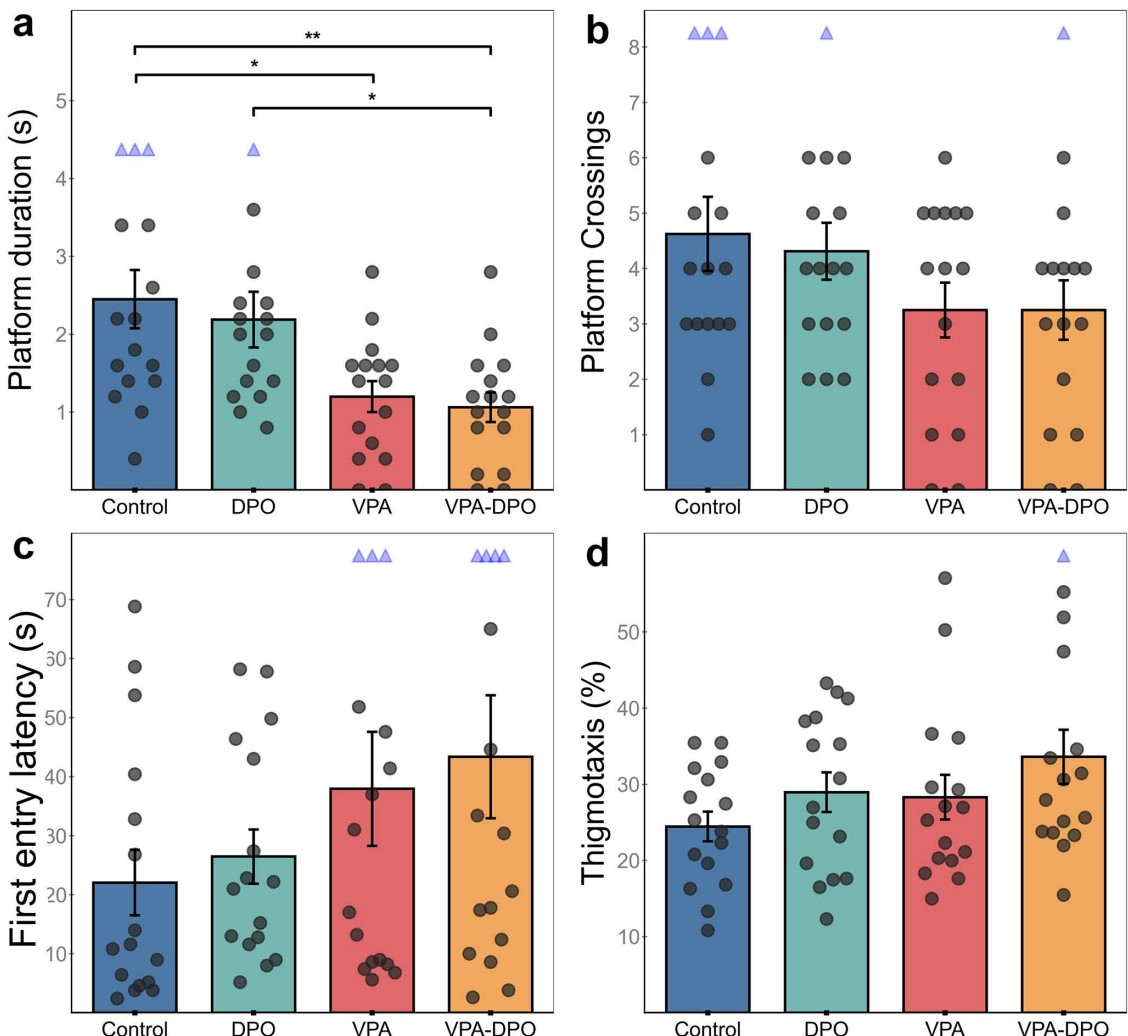

**Fig 5. Morris water maze performance during the probe trial (Day 5).** Each panel displays individual values and group means ± SEM for the four experimental groups. **(a)** VPA and VPA-DPO groups spent significantly less time in the former platform location compared to Control ($p = 0.049$, $d = 1.04$; $p = 0.01$, $d = 1.17$), suggesting impaired spatial memory. **(b)** VPA exposure significantly reduced the number of platform crossings in two-way ANOVA ($F = 4.77$, $p = 0.03$), though post hoc comparisons did not reach significance. **(c)** A main effect of VPA was observed ($F = 4.24$, $p = 0.044$), with delayed initial approaches in both VPA groups; however, pairwise comparisons were not statistically significant. **(d)** No significant group differences emerged in thigmotaxis, though VPA-DPO animals spent more time near the periphery, consistent with patterns observed during training and reversal phases. Statistical comparisons: Group differences assessed using Kruskal-Wallis with Dunn's post hoc and Holm-Bonferroni correction, main effects reported using two-way ANOVA. *$p < 0.05$, **$p < 0.01$. Extreme data points exceeding display limits are shown as upward blue triangles; all values were included in statistical analyses.

findings are presented in the main text. Complete raw data and full statistical results for all variables and days are available in the Zenodo repository. All statistical data for the MWM tests are provided in the Supporting Information S2 and S3 Files.

Representative swim paths (Fig 6) and group-level average occupancy heatmaps (Fig 7) illustrate spatial search patterns on Days 4, 5, and 8, corresponding to the end of initial training, the probe trial, and the conclusion of reversal learning, respectively. These visualizations complement the quantitative analyses but are not included in statistical comparisons.

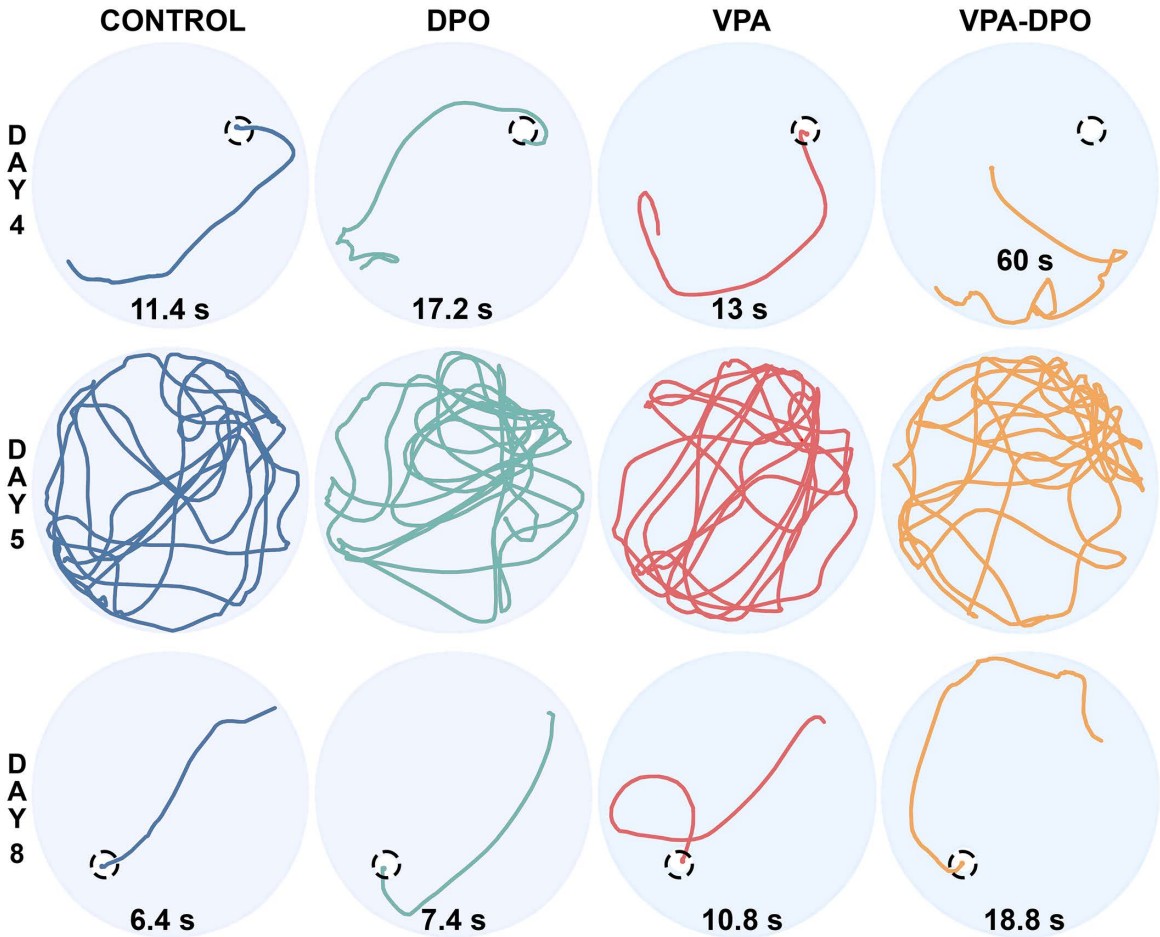

**Fig 6. Morris water maze swim paths.** Representative swim paths were selected as trials that began from the quadrant opposite the platform location (days 4 and 8) and had total distance closest to the group mean. Dashed black circles mark the hidden platform location on day 4 (end of acquisition) and day 8 (end of reversal). On probe day, the platform was removed and animals were allowed to explore for 2 minutes. Escape latency is noted for each trial.

## Hematological parameters

Analysis of hematocrit levels at P28 revealed mean values of 72% (DPO) and 69% (VPA-DPO) compared to 41% (Control) and 31% (VPA). Two-way ANOVA showed a significant main effect of DPO treatment ($F=80.28$, $p<0.0001$) with no significant effect of VPA ($F=3.22$, $p=0.088$) or VPA×DPO interaction ($F=0.61$, $p=0.44$). Post-hoc comparisons showed that DPO significantly elevated hematocrit compared to Control ($p=0.022$, $d=3.38$), and VPA-DPO was significantly higher than VPA ($p=0.009$, $d=3.93$). Analysis of within-animal hematocrit change from P21 to P28 showed that the VPA-DPO group had a significantly greater increase compared to Control ($p=0.013$, $d=2.26$). Analysis of hemoglobin levels at P28 showed a similar pattern, with two-way ANOVA revealing significant main effects of both DPO ($F=99.4$, $p<0.001$) and VPA ($F=5.7$, $p=0.027$), with no significant interaction. DPO treatment elevated hemoglobin to ~23 g/dL (DPO: 24.0, VPA-DPO: 22.5) compared to controls (15.7 g/dL), while VPA exposure alone was associated with lower hemoglobin levels (12.9 g/dL vs 15.7 in controls). Hematocrit levels are shown in Fig 8. Complete statistical data including hemoglobin levels are provided in Supporting Information S4 File.

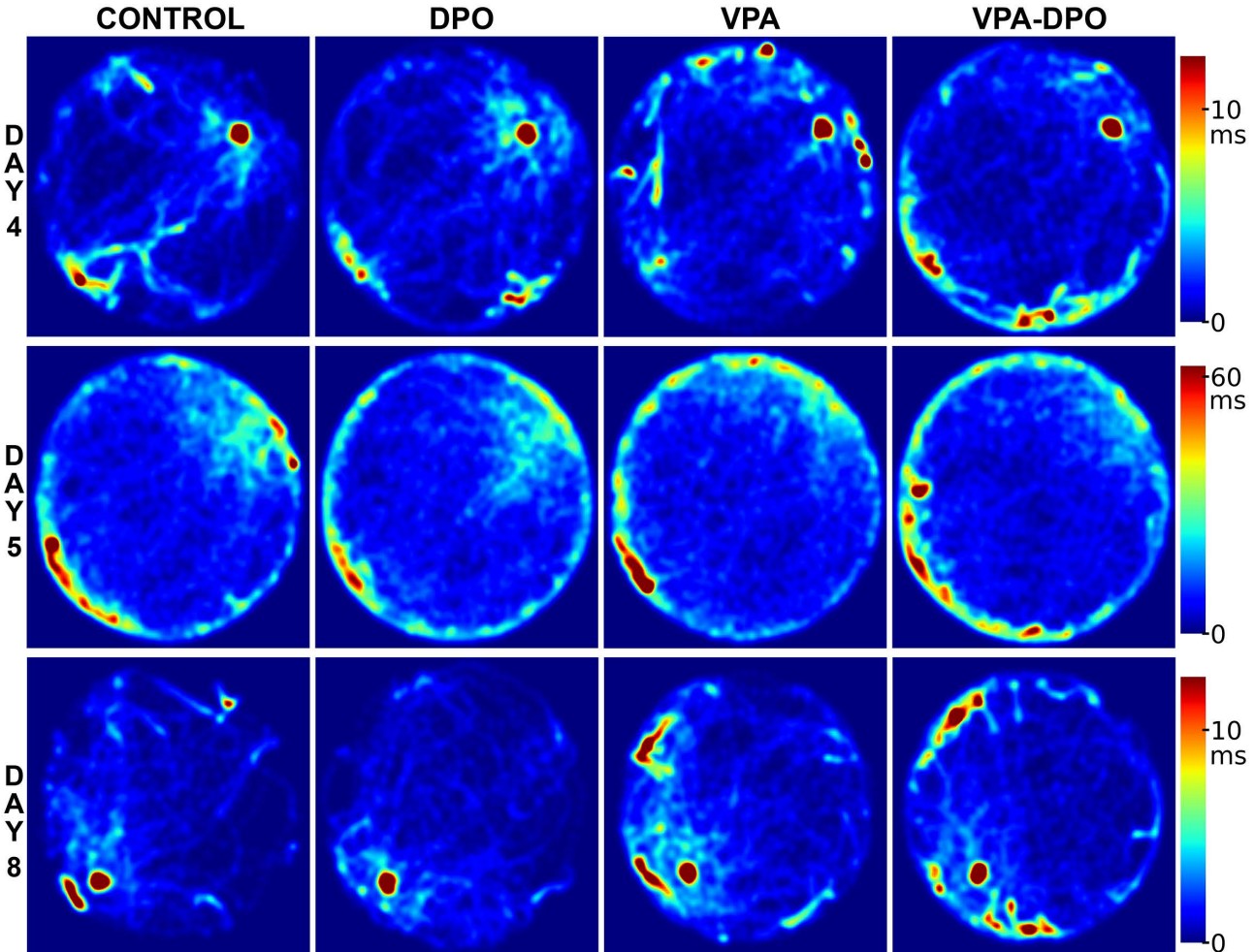

**Fig 7. Morris water maze spatial patterns.** Group-level occupancy heatmaps generated from center-point tracking data depict spatial patterns at the end of acquisition (day 4, platform upper-right), on probe trial (day 5, platform removed), and at the end of reversal learning (day 8, platform lower-left). Each bin corresponds to ~1 cm², color scale indicates time spent per bin. Day 5 uses a different color scale due to longer trial duration (120 s).

## Discussion

We investigated whether postnatal treatment with DPO, a long-acting EPO analogue, could improve autism-relevant behaviors in a VPA rat model at an age roughly aligned with human ASD diagnosis. VPA-exposed animals showed robust impairments in social behavior and spatial learning, consistent with previous studies [30,39]. DPO reliably elevated hemoglobin, confirming systemic activity, but did not ameliorate social or spatial deficits. VPA exposure was associated with reduced hemoglobin, though this was largely overridden by DPO, which elevated hemoglobin to supraphysiological levels regardless of prenatal exposure.

### Behavioral findings

In the three-chamber test, VPA-exposed animals exhibited significant social preference deficits, and DPO treatment did not ameliorate these impairments. On scatter plots, VPA-exposed rats appeared more heterogeneous in their social preference responses. To quantify this apparent dispersion, we conducted Levene's test, which confirmed a significant

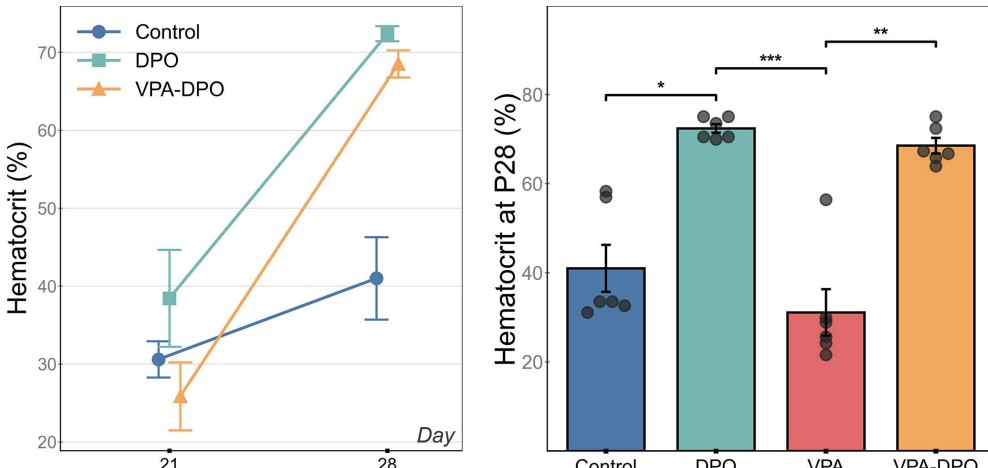

**Fig 8. Hematocrit levels. (a)** VPA-DPO animals showed a significantly greater increase in hematocrit from P21 to P28 compared to Control ($p = 0.013$, $d = 2.26$). **(b)** Hematocrit at P28 was significantly elevated in DPO compared to Control ($p = 0.022$, $d = 3.38$). VPA-DPO was significantly higher than VPA ($p = 0.009$, $d = 3.93$). Two-way ANOVA revealed a significant main effect of DPO ($F = 80.3$, $p < 0.0001$) with no effect of VPA or VPA×DPO interaction. Individual data points shown with bars representing group means ± SEM. Statistical comparisons: (a) one-way ANOVA with Tukey's HSD; **(b)** Kruskal-Wallis with Dunn's post hoc and Holm-Bonferroni correction, two-way ANOVA for main effects. *$p < 0.05$, **$p < 0.01$, ***$p < 0.001$. Note: P21 data for the VPA group was unavailable due to equipment failure.

increase in within-group variability in the VPA cohort. Such phenotypic heterogeneity has been previously reported in VPA rodent models [40,41] and may reflect the behavioral diversity observed in clinical ASD populations. In contrast, the DPO and VPA-DPO groups did not differ significantly from controls in variability. However, this variability effect was not consistent across all behavioral parameters, and therefore our interpretation remains limited. Additionally, VPA-exposed animals showed reduced alternation rates and increased immobility in two-way comparisons, which can be interpreted as proxies for restricted and repetitive behavior domains in ASD. DPO treatment did not significantly modify these secondary behavioral measures.

In the Morris Water Maze, DPO failed to improve the profound impairments in spatial learning, memory retention, and cognitive flexibility observed in VPA animals during both the acquisition and reversal phases. In MWM tasks, DPO-treated animals often reached the platform as quickly as controls but failed to remain on it long enough to register as an escape, leading to prolonged escape latencies during initial training. This discrepancy resolved by the reversal phase, yet persistent thigmotaxis was observed across all phases. These patterns indicate that DPO-treated animals were capable of accurate spatial localization but showed an alteration in strategy or motivation. Similar dissociations between spatial performance and goal-directed execution have been observed in other studies [42].

A nuanced aspect of the MWM results was the complex swim velocity pattern. On earlier days, poorer-performing animals spent more time at steady cruising speeds during extended searches, while successful animals spent proportionally more time accelerating and decelerating, reducing their average velocity. As performance improved and trials shortened, these transient phases became a larger fraction of total time for all groups. By reversal, velocity more accurately reflected sustained strategy, with VPA animals swimming more slowly, likely reflecting altered cognitive flexibility.

## Hematological response

Recent studies have examined EPO's protective effects in propionic acid (PPA)-induced models of autism, reporting improvements following immediate post-insult EPO administration [43,44]. However, these models primarily simulate acute neuroinflammation and function more as neurotoxicity paradigms without addressing developmental treatment

timing. Neither these studies nor the previous work in a developmental ASD model [17] reported hematological outcomes of erythropoietic analogues, leaving it unclear how efficacy relates to hematopoietic load for direct comparison with our findings.

A central finding of our study was the prominent hematological response, with mean hematocrit in DPO-treated animals reaching ~70% compared to control's 41%. A few studies have examined the relationship between neuroprotective dosing of erythropoietic analogues and hematologic effects. Wang et al. [28] administered 35,000–70,000 IU/kg total EPO over 7 days after experimental stroke in adult rats and observed functional recovery alongside hematocrit approaching ~70%, which returned toward baseline within weeks. Cherian et al. [27] tested single-bolus DPO across 2.5–50 μg/kg in adult rats with traumatic brain injury, reported robust histological and neurological recovery with 25 and 50 μg/kg, and hematocrit levels averaging 51% at 2 weeks post-treatment (not dose-stratified). Our stronger hematocrit rise likely reflects the split 5-day regimen (vs single bolus), developmental stage (juvenile vs adult), our data being specific to 50 μg/kg, and critically, measurement timing (3 days post-treatment, likely near peak, vs 2 weeks in Cherian during decline). Viewed in this context, substantial yet transient hematological elevations are compatible with neuroprotective efficacy when intervention timing is optimal. Importantly, our DPO-treated animals remained clinically active throughout the 6-week study and completed all behavioral testing without impairment.

### Therapeutic implications

The existing literature and our findings emphasize the importance of treatment timing as a determinant of neurobehavioral outcome. Treatment at P21, though translationally relevant to typical ASD diagnosis age, may be too late to meaningfully alter established neurodevelopmental trajectories. Haratizadeh et al. [17] demonstrated behavioral improvements with equivalent EPO dosing (2000 IU/kg/day × 5 days) administered at P1-5 in the VPA model, while we observed no benefit at P21-25 despite clear systemic drug activity. Although small, non-significant trends toward improvement may suggest some residual biological activity, the magnitude was insufficient for behavioral efficacy.

While it remains theoretically possible that higher or prolonged dosing could extend the therapeutic window, the pronounced hematological response we observed, combined with clinical evidence of increased thrombotic risk with high-dose EPO in neonates [45] indicates that such dose escalation requires caution, especially when translating these therapies. A recent study using the experimental non-erythropoietic analogue carbamoylated EPO (CEPO) in adult BALB/cJ mice reported improved social behavior [46]. While this strain exhibits inherently low sociability, it remains unclear whether this effect represents a true reversal of autism-relevant deficits in a neurodevelopmental framework [47]. Nevertheless, by eliminating the hematopoietic liability, non-erythropoietic analogues could enable testing of whether intensive dosing can compensate for delayed intervention timing, or alternatively, confirm that the therapeutic window closes regardless of dose optimization.

### Limitations

While autism is currently diagnosed through behavioral criteria, translational research benefits from multimodal assessment approaches. Our behavioral assessments focused on social preference and spatial cognition and were complemented by systemic hematological monitoring, but did not address additional ASD-relevant domains such as communication or sensorimotor processing, nor did we include dedicated assays for restricted and repetitive behaviors (though alternation rate may have provided indirect information). Additionally, we did not include histological or molecular analyses of brain tissue. This study was limited to male rats and to the valproic acid model, which represents only a subset of ASD etiology. Therefore, our conclusions about the therapeutic timing of EPO analogues should be interpreted within these experimental parameters. These limitations highlight directions for future research, including broader behavioral phenotyping, tissue-level mechanistic studies, sex-comparative analyses, and evaluation across alternative developmental models.

## Conclusions

In this study, we investigated whether DPO could improve autism-relevant behaviors in a VPA rat model when administered at a stage corresponding to the human post-diagnostic period. Despite clear systemic activity, DPO treatment failed to produce significant behavioral improvements, suggesting that intervention at this timepoint may be too late to meaningfully alter established neurodevelopmental trajectories. This highlights a central challenge in translating neuroprotective interventions for ASD: agents effective in early or acute windows may not retain efficacy at clinically accessible developmental stages. The pronounced hematological response observed with erythropoietic EPO analogues constrains dose escalation in post-diagnostic windows, though non-erythropoietic derivatives could enable testing of whether intensive dosing can overcome timing limitations or whether the therapeutic window closes absolutely. These results suggest that the treatment efficacy observed in optimal windows does not necessarily predict outcomes at clinically realistic timepoints for neurodevelopmental disorders, highlighting a key consideration for translational study design.

## Supporting information

**S1 File. Three chamber test statistical data.**
(XLSX)

**S2 File. Morris water maze test progression statistical data.**
(XLSX)

**S3 File. Morris water maze test day based statistical data.**
(XLSX)

**S4 File. Hematological parameters statistical data.**
(XLSX)

## Acknowledgments

This manuscript is based on the first author's PhD thesis project. We acknowledge Dr. Marco Canepari and Dr. Bilal Çiğ for insightful discussions during the project and for their comments on the manuscript before submission. AI tools were used to assist with grammar and clarity; all content was reviewed and approved by the authors.

## Author contributions

**Conceptualization:** Ömer Yusuf İpek.

**Data curation:** Ömer Yusuf İpek.

**Formal analysis:** Ömer Yusuf İpek.

**Funding acquisition:** Nurcan Dursun.

**Investigation:** Ömer Yusuf İpek, Tuğba KIRBOĞA.

**Methodology:** Ömer Yusuf İpek, Ercan Babur, Cem Süer.

**Project administration:** Nurcan Dursun.

**Software:** Ömer Yusuf İpek.

**Supervision:** Nurcan Dursun.

**Validation:** Ömer Yusuf İpek, Ercan Babur.

**Visualization:** Ömer Yusuf İpek.

**Writing – original draft:** Ömer Yusuf İpek.

**Writing – review & editing:** Nurcan Dursun, Cem Süer.

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
