## [Decision Letter · Decision Letter 0]

20 Aug 2025

Therapeutic Timing Limitations of Postnatal Darbepoetin in a Valproic Acid Rat Model of Autism Spectrum Disorder

PLOS ONE

Dear Dr. İpek,

Please elaborate on your choice to use darbepoetin instead of EPO for your studies, especially since because of its pharmacokinetics darbepoietin is known to induce a robust increase in red cell production and hemoglobin.  Also describe and discuss more detailed the reasons for the selection of dose of darbepoietin and the time window of treatment as well as the potential consequences of these decisions for the study outcome.Please explain in more detail the pharmacological differences between EPO and darbepoetin.  Please also discuss more in detail the safety profile of EPO in comparison to its analogues.Please also elaborate more in detail in the introduction the advantages and disadvantages of your animal model for autism spectrum disorders (ASD).The details of the behavioral paradigms should be more elaborated in the methods section.  Also please describe the relevance of the behavioral tests for your disease model.Discussion on sex-specific effects is not clearly justified if only male mice were tested.Pay attention to the correct labelling of all axes and legends in the figures.Please add all post hoc comparisons following the two-way ANOVAs, perhaps in a table or a supplementary file.

We look forward to receiving your revised manuscript.

Kind regards,

Anna-Leena Sirén

Academic Editor

PLOS ONE

Journal Requirements:

[This manuscript is based on the first author's PhD thesis project, funded by Erciyes University Scientific Research Projects Coordination Unit]. 

[This manuscript is based on the first author's PhD thesis project, funded by Erciyes University Scientific Research Projects Coordination Unit. We acknowledge Dr. Marco Canepari and Dr. Bilal Çiğ for insightful discussions during the project and for their comments on the manuscript before submission. AI tools were used to assist with grammar and clarity; all content was reviewed and approved by the authors.]

[This manuscript is based on the first author's PhD thesis project, funded by Erciyes University Scientific Research Projects Coordination Unit]

Reviewers' comments:

Reviewer's Responses to Questions

**Comments to the Author**

1. Is the manuscript technically sound, and do the data support the conclusions?

Reviewer #1: Partly

Reviewer #2: Yes

2. Has the statistical analysis been performed appropriately and rigorously?

Reviewer #1: Yes

Reviewer #2: Yes

3. Have the authors made all data underlying the findings in their manuscript fully available?

Reviewer #1: Yes

Reviewer #2: Yes

4. Is the manuscript presented in an intelligible fashion and written in standard English?

Reviewer #1: Yes

Reviewer #2: No

Reviewer #1: The authors present a first-of-its kind study of darbapoetin alfa, an erythropoietin derivative, on autism-related behavior phenotypes in the valproic acid models in a development context. The authors are driven by the observation that erythropoietin and its derivatives seem to be able to correct cellular dysfunctions that appear to play a role in autism and wish to explore this further. The authors correctly point out that previous studies used models with important limitations or did not look at the chronic importance of erythropoietin and its derivatives at a developmentally relevant time point, and the experiments seem to be well-conducted and well-analyzed, but this study also has key important limitations that are under-discussed and affect the overall impact of the paper. The authors find that at a particular dosage of darbapoetin for five days starting on PD21 has very slight impacts on social behaviors and minor impacts on performance in the Morris water maze. These results suggest that this particular dosage scheme has no chronic impacts on the narrow range of ASD-related behaviors tested. The authors explain these null results by suggesting that appropriate developmental time period has passed (though they buffer this remark by pointing out that other erythropoietin drug experiments don't suffer from this problem), that there were effects by increased hematopoietic load, or that that are vague issues regarding baseline sociability or receptor targeted. However, there may be a simpler explanation: the dosage is simply not large enough or has not been delivered for long to have the effects that they wish to rescue, but that the dosage is already too high to be safely administered and it would be unsafe to increase the dose to test this hypothesis. This explanation is not clearly explored in the paper, and any consideration of this finding should lay this out more clearly while also admitting the extremely limited nature of the finding.

Specific comments:

(1) In the last sentence of the abstract, the authors suggest that the therapeutic window for EPO analogues may close earlier, but for the reasons I lay out above, and using evidence they lay out in the discussion, this statement is not supported by the rest of the paper. This should be changed.

(2) In the abstract, and at other locations in the paper, the authors raise the point that EPO analogues may be unsafe for a juveniles. This is not a novel observation--there is a lot of literature about the safety issues surrounding EPO at dosages required for neuopsychiatric differences, and these findings have driven the search for safer alternatives that yield similar results. The authors should do a more careful review of this literature and use this to provide context for their results.

(3) In the introduction, the authors spend time talking about sex differences despite the fact that only males are used in the their paper. They should remove these statements.

(4) In the introduction, the authors should point out the limitations of using VPA to represent the broader autism patient population.

(5) In the introduction, the authors should spend slightly more time talking about the differences between DPO and EPO.

(6) How dose the dosage actually compare to the two studies referenced in the introduction? Because a key limitation to this study is the choice of the dosage itself, the authors should spend more time talking about their choices with specific references to studies where DPO has made a difference neurobehaviorally.

(7) In the methods section, there is a suggestion that testing spanned three consecutive days, but there are ony two experimental days mentioned. Please correct this.

(8) In the results, there is a reference to experiments (a "subsequent novelty trial" in the three-chamber that are not described in the methods. These experiments should be described there, and their findings should be included in the manuscript (perhaps in a supplemental figure). The way these findings are discussed in the methods is not appropriate and they should be described completely. The authors should also talk about why they decided to describe variability.

(9) In the results, the authors use unusual terminology to describe the social and non-social objects in phase 2 of the experiment. Rather than "intruder" and "familiar" they should use other language.

(10) In the results, the authors spend a bit of time speculating as to why the Morris water maze results turned out the way it did. This should be moved to discussion.

(11) The day-specific comparisons should either be described in their entirety or removed.

(12) In the discussion, the authors should thoroughly address the critiques I outline at the beginning of this critique.

(13) In the discussion, the authors should address how these findings compare to other neurobehavioral findings with DPO.

(14) In all of the figures, the y-axes should be labeled appropriated throughout.

(15) For clarity, it would be useful for the authors to report all post hoc comparisons following the two-way ANOVAs, perhaps in a table or a supplementary file.

Reviewer #2: This manuscript examines whether early postnatal darbepoetin alfa (DP; an erythropoietin analogue) mitigates behavioral alterations in offspring exposed in utero to valproic acid (VPA; 2-propylpentanoic acid), a teratogen with relevance to neurodevelopment. The authors evaluate sociability using the three-chamber social approach test and learning/memory using the Morris water maze (MWM), analyze data primarily by ANOVA, and conclude that DP administered at 10 µg/kg/day on postnatal days (PND) 21–25 does not rescue VPA-induced behavioral abnormalities. Overall, the study provides a clear negative preclinical result at the tested dose and time window. Several issues, however, limit interpretability and generalizability.

Major Comments

1. Model validity and scope of inference

The prenatal VPA rat is a well-established, widely used environmental/teratogenic ASD-relevant model. Please specify gestational day(s), dose, and strain, and discuss how these parameters align with canonical protocols, as outcomes are highly timing- and dose-sensitive.

Because ASD is heterogeneous, the manuscript should avoid general conclusions about “ASD” per se. Consider framing results as domain-specific within this model. If feasible, discuss (or acknowledge as a limitation) how findings might compare with complementary models (genetic or immune-activation), which could strengthen mechanistic inference.

2. Rationale for DP regimen and pharmacology

Provide a stronger justification for the dose, route, and timing (PND21–25). Is this window expected to influence circuits implicated by prenatal VPA exposure?

Include, if available, pharmacokinetic or pharmacodynamic markers (e.g., hematocrit/reticulocytes, EPO-receptor pathway readouts, brain exposure) to confirm target engagement.

Discuss literature on earlier neonatal or perinatal EPO-family interventions and whether an earlier window might be necessary.

Clarify whether sex-specific effects were considered; if both sexes were studied, analyze sex as a factor or justify pooling.

3. Behavioral battery adequacy for ASD-relevant domains

The combination of three-chamber sociability and MWM addresses social approach and spatial learning but does not cover other core ASD-relevant dimensions. Please consider (or acknowledge as limitations) assays of:

Restricted/repetitive behavior (e.g., self-grooming scoring, marble burying, stereotypy quantification).

Communication (e.g., juvenile/adult ultrasonic vocalizations).

Sensory processing and sensorimotor gating (e.g., acoustic startle/PPI).

**Do you want your identity to be public for this peer review?** For information about this choice, including consent withdrawal, please see our Privacy Policy

Reviewer #1: No

Reviewer #2: No

---

## [Author Response · Author response to Decision Letter 1]

13 Oct 2025

Dear Reviewers,

Thank you for your thorough and constructive feedback on our manuscript "Therapeutic Timing Limitations of Postnatal Darbepoetin in a Valproic Acid Rat Model of Autism Spectrum Disorder" (PONE-D-25-35324). Your insightful comments have substantially improved the scientific rigor, clarity, and interpretive framework of our work.

We have carefully addressed each point raised in your reviews. The major improvements include:

Enhanced rationale and contextualization:

Comprehensive justification for the P21-25 treatment window, with detailed neurodevelopmental milestones explaining why this period targets synaptic pruning processes relevant to ASD

Expanded discussion of darbepoetin selection, dose calculation based on EPO:DPO bioequivalence ratios, and alignment with published neuroprotective ranges

Thorough review of comparable studies showing neuroprotection at similar hematocrit levels, properly contextualizing our hematological findings

Discussion of non-erythropoietic EPO analogues as promising alternatives for post-diagnostic intervention

Improved model characterization:

Explicit acknowledgment of VPA model advantages (construct validity, established protocols) and limitations (environmental-insult model, male-only cohort, incomplete behavioral domain coverage)

Domain-specific framing of results rather than general claims about ASD

Clear specification of gestational timing (E12.5), dose (500 mg/kg), and alignment with canonical protocols

Methodological transparency:

Complete description of all behavioral tests, including the previously missing social novelty phase

All statistical analyses provided in supplementary files (S1-S4) with complete post-hoc comparison matrices

Clearer interpretation:

Revised abstract and discussion to appropriately weigh evidence regarding therapeutic timing versus hematological constraints

Moved speculative content from Results to Discussion

Improved terminology (social/non-social instead of intruder/familiar)

All figures updated with proper axis labeling and PLOS ONE formatting

All changes are marked with Track Changes in the revised manuscript for easy identification. We provide detailed point-by-point responses to each comment in the accompanying response letter, with specific page numbers for locating revisions.

We believe your feedback has resulted in a significantly stronger manuscript that makes a clearer contribution to understanding the translational challenges of EPO-based therapies for neurodevelopmental disorders. We are deeply grateful for the time and expertise you invested in improving our work.

Thank you for your consideration of our revised manuscript.

Sincerely,

Ömer Yusuf İpek, PhD

(On behalf of all authors)

Department of Physiology, Faculty of Medicine

Ahi Evran University, Kırşehir, Turkey

---

## [Editor Report · Decision Letter 1]

6 Nov 2025

Therapeutic Timing Limitations of Postnatal Darbepoetin in a Valproic Acid Rat Model of Autism Spectrum Disorder

PONE-D-25-35324R1

Dear Dr. İpek,

We’re pleased to inform you that your manuscript has been judged scientifically suitable for publication and will be formally accepted for publication once it meets all outstanding technical requirements.

Kind regards,

Anna-Leena Sirén

Academic Editor

PLOS ONE
---

## [Editor Report · Acceptance letter]

PONE-D-25-35324R1

PLOS ONE

Dear Dr. İpek,

I'm pleased to inform you that your manuscript has been deemed suitable for publication in PLOS ONE. Congratulations! Your manuscript is now being handed over to our production team.

Kind regards,

on behalf of

Dr. Anna-Leena Sirén

Academic Editor

PLOS ONE